# Group Triple P Intervention Effects on Children and Parents: A Systematic Review and Meta-Analysis

**DOI:** 10.3390/ijerph19042113

**Published:** 2022-02-13

**Authors:** Sandra Nogueira, Ana Catarina Canário, Isabel Abreu-Lima, Pedro Teixeira, Orlanda Cruz

**Affiliations:** 1Faculty of Psychology and Education Sciences, University of Porto, 4200-135 Porto, Portugal; sandracat27@gmail.com (S.N.); anacanario@fpce.up.pt (A.C.C.); isabelmpinto@fpce.up.pt (I.A.-L.); 2School of Medicine, University of Minho, 4710-057 Braga, Portugal; pedroteixeira@med.uminho.pt

**Keywords:** evidence-based parenting interventions, parenting support, Group Triple P, Triple P system, level 4 intervention, children’s behavior, parents’ outcomes, systematic review and meta-analysis, risk of bias

## Abstract

Supporting parents through the delivery of evidence-based parenting interventions (EBPI) is a way of promoting children’s rights, given the known benefits to child development and family wellbeing. Group Triple P (GTP) is an EBPI suitable for parents of children aged 2–12 years, who experience parenting difficulties, and/or child behavior problems. Even though GTP has been intensively studied, information lacks on the magnitude of its effects, considering the risk of bias within and across prior research. To address this, a systematic review and meta-analysis (PROSPERO registration CRD42019085360) to evaluate the effects of GTP on child and parent outcomes at short- and longer-term was performed. Through a systematic search of a set of databases, 737 research papers were identified, and 11 trials were selected. The risk of bias within and across studies was evaluated. Significant positive effects of GTP were found immediately after the intervention for child behavior problems, dysfunctional parenting practices, parenting sense of competence, psychological adjustment, parental stress levels, conflict, and relationship quality. Six months after the intervention, positive effects were found only for child behavior problems. Data suggest that GTP might be an effective EBPI leading to positive family outcomes. Substantial risk of bias was found, highlighting the importance of improving the quality of research.

## 1. Introduction

Parents’ behaviors shape the children’s physical, social, and emotional environment and affect their development. Children who experience responsive and consistent parenting, boundaries and contingent limits, and low-conflict family environments, have lifelong advantages such as secure attachment, physical and mental health, higher academic achievement, reduced risk of substance abuse, and reduced risk of antisocial behavior [1,2]. In contrast, children who endure adverse family experiences, such as parental stress, low psychological adjustment, or dysfunctional parenting practices (laxness and over-reacting behaviors) are at increased risk of developing severe psychopathology [1]. Parenting support, as a strategy to promote positive behaviors of parents towards their children, is acknowledged as a child’s right, contributing to the positive development of children and family wellbeing. As a construct, parenting support refers to any activity or intervention targeting parents aiming to reduce risks and enhance protective factors for the children, in relation to their social, physical, and emotional wellbeing [3].

In the last decades, evidence-based parenting programs have proliferated, supported by research documenting that parenting interventions might improve parent-child relationships, promote children’s mental health, prevent child maltreatment, and enhance parental psychosocial wellbeing [4].

The Triple P system combines a set of evidence-based parenting programs according to a public health approach that aims to reach large segments of the population in a supportive, normative, and non-stigmatizing way, allowing access to any family through a stepped-care strategy of five intensity levels, which promotes positive parenting behaviors and prevents, or treats, child behavior problems [2,5]. Triple P is a behavioral family intervention based on a theoretical framework drawing from social learning theory models, research in child and family behavior therapy, developmental research on parenting in everyday contexts, research from the field of developmental psychopathology, social information processing models, and population health research on changing health risk behaviors [6].

Triple P interventions can be delivered in different formats (group, individual, self-directed), targeting specific groups (e.g., Indigenous Triple P [7], Stepping Stones for children with developmental disabilities [8]) and specific problems (e.g., Workplace Triple P [9], Transitions Triple P [10]). The interventions are part of a 5-level system where the intensity increases as the interventions move from universal (level 1) to target (level 5) status, according to the parents’ needs. Level 1 is a universal communication strategy for all interested parents with useful information about parenting. Level 2 is a brief primary health care intervention that provides developmental guidance to parents. Level 3 is a 4-sessions intervention that includes active skills training for parents of children with mild to moderate behavioral problems. Level 4 is an intensive 8-to-10 sessions group or individual parent-training program for parents of children with behavior difficulties. Level 5 is an enhanced behavioral family intervention for families who, beyond parenting difficulties, face additional family stressors (e.g., child maltreatment, marital conflict, parent depression). Normally, level 4 is a mandatory baseline for most families who require a more intensive level 5 intervention.

Triple P interventions aim to prevent emotional, behavioral, and developmental problems in children by enhancing the knowledge, skills, and confidence of parents, emphasizing the outgrowth of self-regulatory capacities in children, parents, and families. Parents learn how to foster children’s development by promoting positive relations with children, encouraging children’s desirable behavior, teaching new skills and behaviors, and managing children’s misbehaviors adequately. A self-regulatory framework is key for parents’ process of change. It means that parents learn to set their own goals, and decide on the parenting skills they want to practice with their children and the tasks to complete between sessions. Parents also learn independent problem-solving skills and are expected to promote independence and autonomy in their children through more positive parent-child interactions. As a result, the interventions foster parental agency and a sense of self-efficacy.

Parenting interventions are commonly offered in a group format, such as the Group Triple P (GTP), a level 4 intervention [2]. In the group format, parents learn specific content on parenting strategies and receive support and constructive feedback from other participants by sharing their problems, which improves their social support networks. Prior reviews have concluded that group interventions contribute to significant improvements in both children’s mental health and parents’ psychological adjustment [11,12,13]. As one of the most widely delivered and studied interventions within the Triple P system [14], GTP requires practitioners/professionals to be trained and accredited, targeting audiences at a selective, preventive, or universal level. GTP can also precede a level 5 intervention in the case of highly stressed families. As such, the spectrum of families’ needs addressed by the intervention is very broad.

Several meta-analyses have been conducted on the effects of Triple P programs. Prior Triple P level 4 meta-analyses evidenced that this level of intervention is particularly effective in decreasing children’s behavior problems and in improving parental practices [15,16,17,18]. Nowak and Heinrichs [19] conducted a Triple P meta-analysis to identify moderator variables for program effectiveness and concluded that better results were associated with a more intensive format, particularly for the families revealing higher levels of distress before the intervention. The systematic review and meta-analysis of multi-level Triple P interventions led by Sanders and colleagues [14] included self-report and observational data and identified significant improvements at short (i.e., immediately after the intervention) and at longer-term (i.e., six and 12-months after the intervention) on children’s behavior problems, parenting practices and sense of competence, parental adjustment and parental relationship.

Despite these positive findings, a few concerns remain regarding the generalization and maintenance of the effects of the interventions and the quality of the research conducted. Wilson and colleagues [20] mention the possibility of selective reporting bias and potential conflicts of interest and conclude that there is no compelling evidence on the effects of Triple P interventions for the whole population or on the maintenance of benefits over time. Sanders and colleagues [14] also identified high or unclear levels of risk of bias in the studies they reviewed. The integrity of the intervention, that is, the degree to which an intervention is delivered as intended, remains an important factor to consider, as it has been associated with better outcomes in the parenting intervention literature [21,22].

Previously published Triple P meta-analyses have focused on one or several levels of intervention and do not present enough evidence on the effects of the GTP level 4 intervention. The purpose of the current review is to: (1) perform a systematic search of primary studies on the effects of GTP, available from scientific and grey literature up to 2020, addressing the quality of the extant research through the evaluation of the risk of bias within and across studies, and (2) estimate the magnitude of effects of GTP on primary and secondary outcomes at short and longer-terms, by comparing the effects of the intervention group with the effects of the control group (waitlist or other non-active interventions). Primary outcomes refer to the variables that GTP targets directly, including child behavior problems (e.g., behavior difficulties, externalizing behavior), dysfunctional parenting practices (e.g., over-reactivity, laxness), and parenting sense of competence (e.g., parents’ problem-solving abilities in their parental role, and satisfaction with parenting). Secondary outcomes include variables that are indirectly affected by interventions, such as parental adjustment, stress levels, parental conflict, and relationship quality.

## 2. Materials and Methods

### 2.1. Protocol and Registration

The study protocol of this systematic review and meta-analysis (PROSPERO, registration number CRD42019085360, available online in https://www.crd.york.ac.uk/prospero/, accessed on 7 February 2022) was prepared according to the Preferred Reporting Items for Systematic Reviews and Meta-Analyses (PRISMA) guidelines [23,24].

### 2.2. Eligibility Criteria

Studies were selected for inclusion if (1) they reported effects of GTP; (2) they were described as individual randomized controlled trials including a GTP intervention condition and a non-active control condition with, at least, pre and post-intervention evaluations; (3) participants were parents or main caregivers of children aged 2–12 years; (4) they included children, and parent or family outcomes; (5) they reported sufficient empirical data for calculating standardized effect sizes; (6) parents did not present severe cognitive impairment; (7) parents and children were not undergoing psychopharmacological treatment; (8) publication language was either English, Spanish, or Portuguese. Studies focusing on cost-effectiveness, acceptability data, practitioner outcomes, consumer satisfaction data, or that reported data from other customized versions of Triple P level 4 were excluded from this review.

### 2.3. Search Methods

Two independent searches were made in each one of the selected databases. No filters were applied. The field “TX All Text” was selected, and the searches included all the available records until 31 December 2020. The first search included the terms “Triple P positive parenting program AND level 4”; and the second search “Group Triple P AND positive parenting program”. The electronic databases searched were: Triple P Evidence Base website, Academic Search Ultimate, CINAHL Plus, Education Source, ERIC, Fonte Académica, MedicLatina, MEDLINE, PsycARTICLES, PsycINFO, Psychology and Behavioral Sciences Collection, American Doctoral Dissertation, Sociology Source Ultimate, Criminal Justice Abstracts, Scopus, Web of Science, PubMed, ProQuest. Furthermore, references from relevant systematic reviews and seminal papers in the field were hand searched. The searches yielded a total of 1633 records. Results for each term search in each database are in Appendix A, Table A1. Figure 1 displays the flow diagram regarding the identification and selection of the studies included in the current review and the procedures are detailed below.

### 2.4. Data Selection and Analysis

From the total of studies retrieved, 869 were duplicated. 737 studies had the titles and abstracts screened by three authors to determine eligibility. 682 studies were excluded for not meeting the eligibility criteria, and 55 full-text articles were assessed for eligibility by the same authors to determine compliance with the inclusion criteria. Any uncertainties regarding eligibility for inclusion were resolved by discussion between the authors. Following the full-text screening, 44 studies were excluded for the following reasons: 35 did not meet the inclusion criteria, seven studies used data from the same samples as previous studies (in these cases, the first published studies were selected), and two studies had no full-text available. Thus, a total of 11 studies was included in the qualitative and quantitative syntheses.

### 2.5. Data Extraction and Management

Eligible studies were reviewed by two independent authors and information was extracted (c.f. Table 1) from each paper on (1) study characteristics, including authors, year and journal of publication, country where the study was conducted, type of control condition, measurement time points, attrition rates; (2) characteristics of trial participants, including setting, sample size, child age and gender, and family socio-economic status; (3) type of intervention, (4) outcome measures, including the instruments used; and (5) developer involvement. Authors were asked to provide the missing information whenever means and/or standard deviations of the pre-intervention, post-intervention, or follow-up scores were not available in published reports.

**Table 1 ijerph-19-02113-t001:** Study characteristics of studies included in the quantitative synthesis.

Authors, Year	Design	Groups	Triple P Intervention/Control Group (I1/I2/C)	Study Approach	Setting	Measurement Time Points	Sample Size (*N*)	Child Mean Age (Range)	% Boys	Developer Involvement	Country	Attrition Rate Post-Intervention I1/C or I1/I2/C	SES	Parent Measures	Child Measures
Au et al., 2014 [25]	RCT	1 I, 1 C	GTP + 1 ADHD booster session/waitlist	Target	clinical (ADHD)	Pre, Post, 3-moFU	17	*7*.81 years (5–10)	94.1	2	Hong Kong	NR	NR	PSOC	ECBI
Bodenmann et al., 2008 [26]	RCT	2 I, 1 C	GTP/CCET/ waitlist	Universal	community	Pre, Post, 6-mo FU, 12-moFU	300	6.6 years (2–12)	55.0	1	Switzerland	4%/20%	medium	PS, PSOC, PPC	ECBI
Chung et al., 2015 [27]	RCT	2 I, 1 C	GTP/DI/waitlist	Universal	community	Pre, Post	88	50.7 months (2–6)	53.3	1	Hong Kong	11%/ 10%/0%	NR	PSS	ECBI
Frank et al., 2015 [28]	RCT	1 I, 1 C	GTP/ waitlist	Universal	community	Pre, Post, 6-moFU	84	5.55 years (3–8)	69.0	1	New Zealand	NR	high	PS, PTC, PPC, RQI	ECBI
Glazemakers, 2012 [29]	RCT	1 I, 1 C	GTP/ waitlist	Treatment	clinical (psychiatric problems)	Pre, Post	50	8.01 (<12)	74.4	2	Belgium	NR	NR	PS, PSI	SDQ
Leung et al., 2003 [30]	RCT	1 I, 1 C	GTP/ waitlist	Target	clinic	Pre, Post	91	4.23 years (3–7)	63.8	1	Hong Kong	28.3%/20%	NR	PS, PSOC, PPC, RQI	ECBI
Leung et al., 2013 [31]	RCT	1 I, 1 C	GTP/ waitlist	Target	clinical (developmental disability)	Pre, Post, 6-moFU	81	49.6 months (NR)	70.4	1	Hong Kong	7.1%/10.3%	NR	PSS, PS, PPC	ECBI
Matsumoto et al., 2007 [32]	RCT	1 I, 1 C	GTP/ waitlist	Universal	community	Pre, Post, 3-moFU	50	4.9 years (2–10)	54.0	1	Australia (Japanese parents)	0 %/0 %	NR	PS, PPC, RQI, PSBC, DASS	ECBI
Matsumoto et al., 2010 [33]	RCT	1 I, 1 C	GTP/ waitlist	Universal	community	Pre, Post	54	5.8 years (2.2–10.3)	NR	1	Japan	10.7%/0%	NR	PS, PPC, RQI, PSBC, DASS	ECBI
Ozyurt et al., 2019 [34]	RCT	1 T, 1 C	GTP/ waitlist	Treatment	clinical (anxiety problems)	Pre, Post	74	9.96 years (8–12)	62.5	2	Turkey	29.7%/21.6%	NR	NR	SDQ
Tully & Hunt, 2017 [35]	RCT	2 I, 1 C	GTP/BPI/ waitlist	Universal	community	Pre, Post, 6-moFU	132	31 months (24–46)	69.6	2	Australia	13%/ 8.3%/9.1 %	NR	PTC, QMI, DASS	CBCL

Note. Design: RCT—Randomized controlled trial; Groups: I1 = intervention 1 group, I2 = intervention group 2, C = control group; Triple P intervention/control group: ADHD = Attention Deficit Hyperactivity Disorder, CCET = Couples Coping Enhancement Training, DI = Brief Parent Discussion Group, BPI = Level 3 Triple P Parent Discussion group; Measurement time points: pre = before the intervention, post = after the intervention, moFU = months follow-up; Developer involvement: 1 = Any developer involvement, 2 = No developer involvement; Attrition rate post-intervention per group: I1/C = intervention group 1/control group, I1/I2/C = intervention group1/intervention group 2/control group, NR = Not reported; SES: NR = Not reported; Parent measures: PSOC = Parenting Sense of Competence, PS = Parenting Scale, PPC = Parent Problem Checklist, PSS = Chinese Parental Stress Scale, PTC = Parenting Tasks Checklist, RQI = Relationship Quality Inventory, PSBC = Problem Setting and Behavior Checklist, DASS = Depression Anxiety Stress Scales, QMI = Quality of Marriage Index, NR = Not reported; Child measures: ECBI = Eyberg Child Behavior Inventory; SDQ = Strengths and Difficulties Questionnaire.

### 2.6. Procedures to Evaluate Risk of Bias within and across Studies

The quality of the selected trials was evaluated based on international guidelines for intervention research, specifically the CONSORT statement [36], the CONSORT-SPI 2018 extension [37], and the Cochrane Collaboration Tool [38]. In the current review, studies were assessed in relation to different categories of bias: (a) random sequence generation; (b) allocation concealment; (c) blinding of participants and personnel; (d) blinding of outcome assessment; (e) reporting of incomplete outcome data; (f) selective reporting of the data; (g) other types of bias (e.g., possible confounding bias and recruitment bias of the participants). Two authors evaluated the risk of bias for each category. The risk was either “high” (i.e., plausible risk of bias that seriously weakens confidence in results), “low” (i.e., a risk unlikely to alter results seriously), or “unclear” (i.e., a plausible risk of bias that raises some doubt about the results). Disagreements were solved through discussion among the authors. The methodological quality of the studies and risk of bias assessment did not interfere with the selection of studies.

### 2.7. Data Analyses and Statistical Approach

To perform a meta-analysis, two models may be considered about the nature of studies and mechanisms of assigning weights, namely the fixed-effect model and the random-effect model. The first model assumes that all studies are identical and that the true effect size is equal for all of them. In the second model, the assumption is that studies are not identical, those different moderators may exist, and there is not a true effect shared by all included studies [39]. For the current analyses, the random-effect size model was selected, and the possibility of different sources of heterogeneity in the sample was assumed. A forest plot was performed, and the relation between sample size and the effect size was analyzed for each outcome variable.

Dependent variables tackled in the studies were classified into primary and secondary outcomes. Primary outcomes included (1) child behavior problems; (2) dysfunctional parenting practices (i.e., laxness and over-reacting behavior), and (3) and parenting sense of competence. Secondary outcomes referred to (4) psychological adjustment (i.e., depression, anxiety, and stress levels); (5) parental stress levels; (6) parental conflict; and (7) relationship quality. Measures corresponding to each outcome category are presented in Table 2.

**Table 2 ijerph-19-02113-t002:** Measures included in each outcome category.

Outcome Variables	Measures
Child behavior problems	Child Behavior Checklist (CBCL) [40]; Eyberg Child Behavior Inventory (ECBI) [41]; Strengths and Difficulties Questionnaire (SDQ) [42].
Dysfunctional parenting practices (total score), laxness subscale, and over-reactivity subscale	Parenting Scale (PS) [43].
Parenting sense of competence	Problem Setting and Behavior Checklist (PSBC) [44]; Parenting Sense of Competence (PSOC) [45]; Parenting Tasks Checklist (PTC) [46].
Parental adjustment, depression levels, anxiety levels, and stress levels	Depression Anxiety Stress Scales (DASS) [47].
Parental stress levels	Parenting Stress Index (PSI) [48]; Chinese Parental Stress Scale (PSS) [49].
Parental conflict	Parent Problem Checklist (PPC) [50].
Parental relationship	Relationship Quality Index (RQI) [51]; Quality of Marriage Index (QMI) [51].

The Review Manager 5.3 software was used to perform separate analyses for each outcome category. Hedges’ G was computed to determine the effect size for this meta-analysis [52]. The primary source for calculating Hedges’ G was the standardized mean difference, as it provides a scale-free estimate of treatment effect that can be compared across different scales as data were extracted exactly as presented in primary studies. When a confidence interval includes zero, it indicates that an effect size is not significantly different from zero. As suggested by Becker [53], and Morris and DeShon [54], the final effect size was intergroup Hedges’ G as the measure of effect size immediately post-intervention, and at six- and 12-month follow-ups, where G represents the standardized mean difference between intervention and control conditions.

Results were presented as effect sizes with a 95% confidence interval. Effect sizes were interpreted as per Cohen’s guidelines [55]; smaller than 0.20 indicated no evidence of effect size, between 0.20 and 0.40 were considered small, between 0.40 and 0.75 were considered moderate, and higher than 0.80 were considered large. The 95% confidence intervals were used to determine significance.

Whenever trials compared GTP with other parenting interventions beyond the waitlist control group, only data comparing GTP with a randomized waitlist control group were considered to ensure that all effect sizes were calculated in reference to comparable groups. The literature is scarce on the comparison of GTP with other group-based parenting interventions, with the work by Lindsay and colleagues [56] comparing GTP with two other group-based parenting interventions being the only one identified. Trials that compared GTP with higher intensity Triple P interventions were identified but considered inadequate for the purposes of the current meta-analysis.

The usual procedure to analyze heterogeneity in a meta-analysis is to conduct subgroup analyses to explore whether findings are consistent across studies involving different samples [57]. The heterogeneity in the subgroups was quantified using the *I^2^* index to detect circumstances under which GTP had consistent effects on all relevant outcome categories. The *I^2^* index stands for the percentage of variation across studies that is due to heterogeneity rather than chance [58]. An *I^2^* statistic of 30% to 60% is frequently interpreted as moderate, whereas an *I^2^* between 50% and 90% indicates high heterogeneity [38]. Any meta-analysis is expected to present some degree of heterogeneity, given that studies with different methods, participants, and measures are combined [58].

## 3. Results

### 3.1. Study Characteristics

Eleven randomized controlled trials were included in the current review. These trials took place in seven countries: New Zealand, Japan, Switzerland, Belgium, and Turkey (one study in each country); Hong Kong (four studies); and Australia (two studies). Ten studies were published within a 16-year period (2003–2019). Six studies published after 2013, not considered on the previous meta-analysis, were included in the current meta-analysis [25,27,28,34,35] as well as an unpublished research work [29].

The developer of GTP was involved in seven of the 11 trials selected [26,28,30,31,32,33]. A total of 885 families were included in the 11 trials. Sample sizes ranged from 17 to 300 (*M* = 105). The mean age of children was 5.2 years, ranging from 2 to 12 years. The mean age of mothers was 37.80 years, ranging from 35.0 to 39.0, and the mean age of fathers was 40.0 years, ranging from 38.0 to 43.0. Four studies did not report the mean age of participants [29,31,32,33]. Most of the participants in the selected trials had a high school or university degree. Five trials with selective or treatment approaches were implemented in clinical settings [25,29,30,31,34], and six studies sharing a universal approach were implemented in community settings [26,27,28,32,33,35].

Seven studies followed the original five group sessions and three telephone sessions format [25,27,28,29,32,33,34], whereas three other studies implemented four group sessions and four telephone sessions [26,30,35]. Leung and collaborators [31] delivered six group sessions with two telephone sessions. All trials compared the intervention group with a non-active intervention group. Three studies evaluated the effects of GTP, comparing the effects of the GTP intervention with those from a different intervention and of a waiting list control group [26,27,35]. Ten trials were published in scientific journals, and one was an unpublished doctoral dissertation [29]. 

Seven studies reported attrition rates ranging from 0 to 29.7% for the intervention group between pre and post-test, whereas three studies did not report attrition rates [25,28,29]. All studies declared to have used the intention-to-treat analysis approach. Ten of the studies included in the systematic review reported data on child sex, with the proportion of parents of boys ranging from 53.3% to 94.1% [25,26,27,28,29,30,31,32,34,35]. In addition, ten trials included parents with typically developing children exclusively, whereas the trial of Leung and colleagues [31] included parents of children with developmental disabilities. None of the selected studies reported data regarding the fidelity of the interventions delivered.

### 3.2. Risk of Bias within and across Studies

The figures depicting the assessment of the risk of bias of the included studies is displayed in Figure A1 and Figure A2 in Appendix B. Six studies did not report how random sequence was generated [25,26,27,29,30,32], and five studies omitted whether allocation sequence was concealed (selection bias) [26,29,30,32,33]. Given the nature of the interventions, blinding of participants is not possible since both practitioners and participants are aware of the type of intervention, either on the delivering or receiving end. As such, all studies revealed a high risk of bias for blinding of participants and study personnel (performance bias). Ten studies were also rated with a high risk of detection bias (lack of blinding of outcome assessment) [25,26,27,28,29,30,31,32,33,35], stemming from the fact that parent report was used as the most common outcome source. Only one study showed a low risk of detection bias [34]. Regarding incomplete outcome data [25,26,27,28,31,32,33] and selective reporting [25,26,27,30,31,32,33,35], eight studies were rated as “low risk”. Finally, two studies were rated as “high risk” for other types of bias, such as possible confounding bias [27,29]. The absence of information regarding Triple P trial registration hindered the assessment of publication bias across studies.

### 3.3. Short-Term Intervention Effects 

All the forest plots from analyses performed for each outcome variable are presented in Appendix C (see Figure A3, Figure A4, Figure A5, Figure A6, Figure A7, Figure A8, Figure A9, Figure A10, Figure A11, Figure A12, Figure A13 and Figure A14). Table 3 displays the effect sizes for seven outcomes at post-intervention. 

Results from the meta-analysis indicate that GTP had a moderate effect size in all primary outcomes. There was a moderate decrease in child behavior problems (SMD: −0.53, 95% CI [−0.71, −0.35], Figure A3) as well as in dysfunctional parenting practices, namely laxness (SMD: −0.46, 95%CI [−0.65, −0.27], Figure A5) and over-reactivity behaviors (SMD: −0.64, 95% CI [−0.83, −0.45]). Regarding parenting sense of competence, GTP also evidenced a moderate increase (SMD: 0.58, 95% CI [0.36, 0.79], Figure A6).

Results regarding secondary outcomes demonstrated a small effect size in decreasing depression (SMD: −0.38, 95% CI [−0.68, −0.09]), and moderate effect size in decreasing stress (SMD: −0.43, 95% CI [−0.73, −0.14], Figure A8) levels, as well as in decreasing parental stress levels (SMD: −0.42, 95% CI [−0.70, −0.13], Figure A9). Small effect sizes were found for GTP on the reduction of parental conflict (SMD: −0.25, 95% CI [−0.48, −0.02], Figure A10) and the improvement of relationship quality (SMD: 0.25, 95% CI [0.01, 0.49], Figure A11). 

### 3.4. Longer-Term Intervention Effects 

Two studies reported data for the six-month follow-up [26,28] and only one study for the 12-month follow-up [26]. Three outcome categories at six-month follow-up were analyzed, and the respective effect sizes are presented in Table 4.

At the six-month follow-up, results on GTP intervention primary outcomes yielded a moderate effect size in decreasing child behavior problems (SMD: −0.53, 95% CI [−0.80, −0.25], Figure A12). Non-significant effect sizes were found for dysfunctional parenting practices total score (SMD: −0.50, 95% CI [−1.03, 0.03], Figure A13) and for parental conflict (SMD: −0.34, 95% CI [−1.00, 0.31], Figure A14).

## 4. Discussion

The current systematic review and meta-analysis addressed the effects of GTP at short and longer-term to compile and analyze updated research to discern if there have been improvements in the quality of recent research (i.e., substantial less risk of bias) posterior to the last published review studies [14,20] as well as any new findings.

The review study included 11 original studies describing GTP trials, with a total of 885 families from seven countries. The literature search went beyond peer-reviewed journals and included dissertation databases and clinical trial websites, providing a comprehensive overview of GTP extant research trials. 

Results suggest that, in the short run and for the primary outcomes under scrutiny, GTP is an effective parenting program. Moderate effect size improvements were found for all GTP targeted outcomes, including child behavior problems, dysfunctional parenting practices (laxness and over-reactivity behaviors), and parenting sense of competence (self-efficacy and parental satisfaction). These findings are consistent with the goals of GTP and are in line with another Triple P intervention review [14]. Small effect sizes were found for all short-term secondary outcomes (psychological adjustment, parental conflict, and relationship quality), except for parental stress levels, which showed a moderate effect size. These findings may be explained as positive side-effects of parenting interventions, whereby the improvements in child behavior problems, parenting practices, and sense of competence bring about more positive and harmonious parenting context, fostering greater familiar wellbeing, better adjustment, less conflict, and better relationship quality among parents.

Regarding the maintenance of intervention effects, findings are limited in that only two trials included data at six-months follow-up [33,54]. Even so, GTP only evidenced a moderate effect size for child behavior problems six months after the intervention, which is somewhat consistent with the findings of a prior systematic review and meta-analysis identifying that parenting interventions lead to sustained effects on child disruptive behavior over time [59]. 

The evaluation of the risk of bias for randomized controlled trials as per CONSORT recommendations draws attention to the specific characteristics of the studies included in the current systematic review and meta-analysis. All studies revealed a high risk of bias for the blinding of participants and performance bias, and all but one revealed a high risk of detection bias. While some of the findings relate to actual shortcomings of the studies and deserve serious consideration, other findings stem from the very nature of the interventions under consideration. Most of the studies included in this meta-analysis did not report important data such as information about randomization and mechanisms used to conceal the allocation and did not specify primary and secondary outcomes beforehand. Additionally, none of the studies included a trial registry number or disclosed a full trial protocol, hindering the assessment of publication bias across studies. Outcome reporting bias was found in two studies, either because not all available subscale scores were reported or because total scale scores were not reported for some of the measures.

Several studies did not present relevant demographic characteristics (child’s sex, parent’s age, and education level, or socio-economic status of the family), which precluded the performance of more complex statistical analyses in the current meta-analysis. In fact, child and parent age, parent education attainment, and socio-economic level should be considered when evaluating the effectiveness of parenting interventions to determine who benefits the most from them. Having such data available would allow for further moderation analysis to clarify some of the unexplained variances. Still, information on the socio-economic status and on families’ risk factors is key to further assessing the effectiveness of the parenting interventions delivered to vulnerable families. Future research is needed to address such gaps and limitations and to inform, from a public health perspective, about strategic decision-making regarding the implementation of GTP.

It also became evident that most GTP research relies solely on parent self-report measures, a practical and feasible way of collecting information on the children and other outcome measures, but certainly a limitation of the generalization of findings and consequently of the present meta-analysis. As such, future primary studies should incorporate independent measure observations of parent and child behavior into their trial designs to provide confirmatory information about the effects of GTP. Noteworthy is the fact that none of the selected studies reported on how the integrity of intervention was monitored, a relevant issue to consider in future research about the effects of GTP and to guarantee the quality and fidelity in the delivery of the programs.

The need for independent research about GTP also becomes evident from the present study. Seven out of the 11 trials included the author’s participation in the Triple P system, and four of them did not declare any conflict of interest. The previous meta-analysis underlined the high risk of bias within primary research [14,20], and the current review did not identify relevant quality improvements among the primary studies published after 2014. In addition, among the 11 trials included, only three trials assessed the longer-term effects, which is manifestly scarce to evaluate and to underpin the efficacy of the intervention. Future research should explore the effects in both the short and longer-term, contributing to strengthening the evidence base of GTP.

## 5. Conclusions

This systematic review and meta-analysis showed that GTP is an effective program in the short term, especially for the primary outcomes that it aims to improve, such as child behavior problems, dysfunctional parenting practices, and parenting sense of competence. However, more research is needed regarding the longer-term effects, which follow participants over time, before conclusions are drawn on the maintenance of GTP efficacy.

Even though the findings of the current systematic review and meta-analysis point out positive effects of GTP, the results also identified different degrees of risks of bias among the studies included in both quantitative and qualitative syntheses, which suggests that the quality of the studies included in the current work may be undermined. A final recommendation includes the need to raise awareness among the scientific community regarding the quality of reporting scientific research of experimental nature, where the adherence to quality standards on the implementation and evaluation of parenting interventions is crucial. The results of sound scientific research on the effects of parenting interventions reinforce and sustain the evidence on parenting interventions and contribute to informing and guiding the decision-making of professionals, stakeholders, policymakers, and parents about the utility, relevance, and usage of evidence-based parenting interventions.

## Figures and Tables

**Figure 1 ijerph-19-02113-f001:**
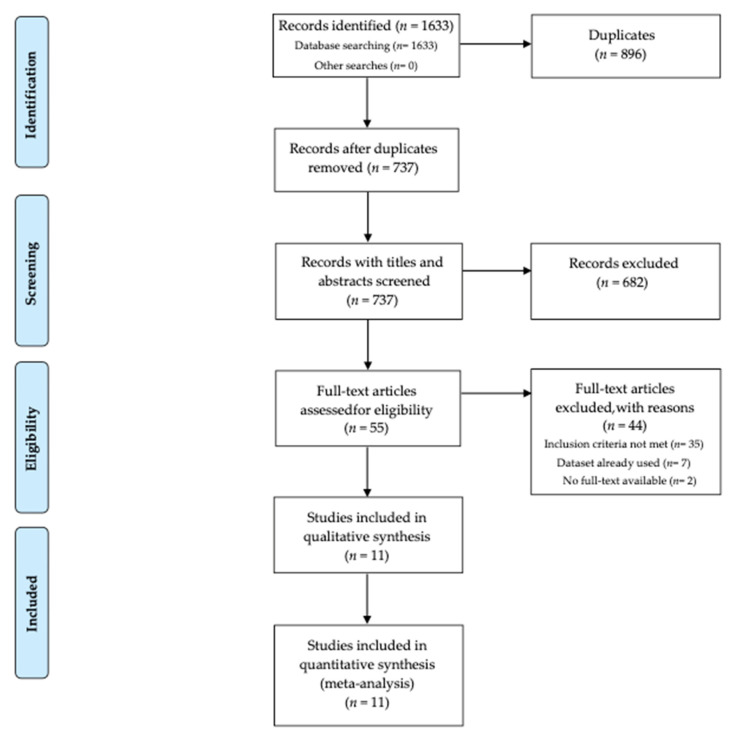
Prisma flow diagram for the studies included in and excluded from the meta-analysis.

**Table 3 ijerph-19-02113-t003:** Effect size for all outcomes categories at post-intervention.

Outcome Categories	*k*	*g*	*d 95%CI*	*z*	*p (for g)*	*Q^2^*	*p (for Q^2^)*	*I^2^*
Primary outcomes								
Child behavior problems	11	−0.53	[−0.71, −0.35]	5.85	0.00	12.50	0.26	20
Laxness subscale (PS)	8	−0.50	[−0.68, −0.32]	5.37	0.00	5.41	0.61	0
Over-reactivity subscale (PS)	7	−0.64	[−0.83, −0.45]	6.53	0.00	2.17	0.90	0
Parenting sense of competence	6	0.58	[0.36, −0.79]	5.19	0.00	5.69	0.34	12
Secondary outcomes								
Depression level (DASS)	3	−0.38	[−68, −0.09]	2.52	0.01	1.48	0.48	0
Anxiety level (DASS)	3	−0.30	[−0.59, 0.00]	1.97	0.05	0.25	0.88	0
Stress level (DASS)	3	−0.43	[−0.73, −0.14]	2.85	0.00	1.38	0.50	0
Parental stress level	4	−0.42	[−0.70, −0.13]	2.87	0.00	0.79	0.85	0
Parental conflict	5	−0.25	[−0.48, −0.02]	2.10	0.04	1.31	0.86	0
Relationship quality	5	0.25	[0.01, 0.49]	2.01	0.04	2.04	0.73	0

Note. *g* = Hedge’s g; *Q* = test statistic for heterogeneity; *k* = number of studies; *p* = test for significance evaluated against 0.05; *I^2^* = measure of degree of heterogeneity.

**Table 4 ijerph-19-02113-t004:** Effects sizes for outcomes categories at 6-months follow-up.

Outcome Categories	*k*	*g*	*d 95%CI*	*z*	*p (for g)*	*Q*	*p (for Q)*	*I^2^*
Primary outcomes								
Child behavior problems	2	−0.53	[−0.91, −0.14]	2.69	0.00	1.87	0.17	46
Dysfunctional parenting practices	2	−0.46	[−0.73, −0.19]	3.37	0.00	0.99	0.32	0
Secondary outcomes								
Parental conflict	2	−0.08	[−0.34, 0.19]	0.58	0.56	0.66	0.42	0

Note. *CI* = confidence interval; *g* = Hedge’s g; *Q* = test statistic for heterogeneity; *k* = number of studies; *p* = test for significance evaluated against 0.05; *I^2^* = measure of degree of heterogeneity.

## Data Availability

The data reported in the study are available in the empirical studies included in the systematic review and meta-analysis.

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
