# Peer review of "Group Triple P Intervention Effects on Children and Parents: A Systematic Review and Meta-Analysis"

_ijerph, 2022, doi:10.3390/ijerph19042113_

Round 1
Reviewer 1 Report
The current manuscript intends to analyse the effectivity of a specific type of intervention, Group Triple P. For this purpose, the authors a metaanalysis in which they review clinical trials testing the outcomes of such kind of intervention. It is remarkable that the study was prerregistered and was performed following PRISMA guidelines. Despite the valuable contribution to the field and clinical practice, some aspects should be addressed before this manuscript is ready for publication.
- Authors should present more clearly the search equation and state additional filters they used in the search.
- Authors should report the results obtained in every search and how the number of potential articles is reduced, indicating the method used for narrowing the search (e.g. what kind of screening) at each stage. This kind of information is more appropiately placed in methods section.
- It is somehow conflicting to find that some of the analyzed articles are focused on clinical populations such as children with ADHD or developmental disability when one of the inclusion criteria was: "participants were not cognitively delayed". Authors should justify or better clarify their decision to include these papers.
- I suggest authors to discuss whether age of parents or children might be a relevant variable to take into account when estimating the effectiveness of the intervention.
Reviewer 2 Report
Thank you for allowing me to review this manuscript. Overall it is very well-conducted systematic review. I have 2 issues, one of which is critical to be addressed and the other optional but would add to the manuscript recommendations for future research.
1) compulsory - Figure 1 and the accompanying text do not match. Also neither the figure nor the text add up properly:
Figure 737 - 639 = 98 not 55 what happened to the other 43 articles
Text 737 to 61 - 49 = 12 not 11. Why only 11 not 12 and also how did authors get to 61 from 737.
This needs to be made clear and text and figure harmonized to correct numbers.
2) Optional - I noted that the parents appeared to be generally older and more educated. No studies are described in lower resource settings. This could be a research recommendation as parenting in middle and low income countries is an emerging issue in global child health.
Reviewer 3 Report
This paper presents a systematic review and meta-analysis on an important topic. The subject is relevant, the aims are clear and you have chosen an appropriate research methodology. I believe you will contribute to our understanding of this important aspect of the short- and long-term effects of GTP on child and parents’ outcomes.
The paper is generally well-written and gives a neat impression. Although I have some minor points the authors should consider, I would recommend accepting this piece of work for publication.
In introduction I miss a paragraph with the theoretical frame of Tiple P.
In line 116 instead of “parental sense of competence” do you mean “parenting sense of competence”? Please give the examples (or define) child behavior problems, dysfunctional parenting practices, and parental sense of competence in line 116.
Table 1 and 2 belong to Results and not Material and Methods.
Limitations of this study and future research areas should be given more thought and expanded upon.
Reviewer 4 Report
This an interesting issue. Congratulations for your work.
Abstract: “Supporting parents through the delivery of evidence-based parenting interventions (EBPI) is acknowledged as a right of children…”
This proposition seems to me polemic. I think that what can be considered as an acknowledged right of children is the right to grow up within a healthy physical, social, and emotional environment, namely with their parents. The EBPI can help but it is not something that must always exist, except when a child is at risk.
Abstract: “Six months after, positive effects were found for child behavior problems.”
It will be better to say that “positive effects were found only for child behaviour problems”. In the discussion section, this fact is not discussed. Why do you think that could be the reason for the absence of other significative results?
Line 108: “The purpose of the current review is to:…”
Lines 143-145: “Search terms included: “Triple P, positive parenting program AND level 4”; “Group Triple P AND positive parenting program”.
It seems that what is presented as the purposes for the study don’t match the chosen terms for the research. The first Boolean sentence has a comma, which is strange. Why do you restrict your search only to level 4? This is not mentioned as an eligible criterium. Please, enhance this explanation of the method.
Final remarks:
I think that the bias encountered in most of the studies could be resumed in the conclusions, because it seems to me a major conclusion of your study, and then, you can present your recommendations as you have done.
I wonder if a period of six or twelve months should be called ‘long-term’. Can you be more precise in relation to the period called ‘short-term intervention effects, which is called “immediately after the intervention” in the abstract? Probably, the effects until the sixth month can all be considered short-term.
